# Can Natural Domain Foundation Models Be Applied to Cardiac MRI Reconstruction?

**Anam Hashmi**[1]                                    ANAM.HASHMI2@MAIL.DCU.IE
[1] *Research Ireland Centre for Research Training in Machine Learning, DCU, Dublin, Ireland*
**Julia Dietlmeier**[2]                              JULIA.DIETLMEIER@INSIGHT-CENTRE.ORG
[2] *Insight Research Ireland Centre for Data Analytics, DCU, Dublin, Ireland*
**Mayug Maniparambil**[2]                            MAYUG.MANIPARAMBIL2@MAIL.DCU.IE
**Carles Garcia-Cabrera**[3]                         CARLES.GARCIA-CABRERA@UCD.IE
[3] *School of Medicine, University College Dublin, Dublin, Ireland*
**Kathleen M. Curran**[3]                            KATHLEEN.CURRAN@UCD.IE
**Noel E. O'Connor**[2]                              NOEL.OCONNOR@INSIGHT-CENTRE.ORG

**Editors:** Accepted for publication at MIDL 2025

## Abstract

The field of computer vision has experienced a paradigm shift with the emergence of general-purpose foundation models, which exhibit strong generalization capabilities across a wide range of tasks. However, their applicability to specialized medical imaging tasks, particularly cardiac MRI reconstruction, remains underexplored. In this work, we investigate the transferability of state-of-the-art vision foundation models like CLIP and DINOv2 for cardiac MRI reconstruction. We propose a novel framework that leverages frozen vision foundation models as image encoders, combined with a UNETR-based trainable decoder. We validate our framework on the CMRxRecon2024 dataset, demonstrating improved performance over the traditional state-of-the-art U-Net under acceleration factor ($\times 4$), despite relying on frozen natural-domain foundation model and significantly fewer trainable parameters. The code will be released at https://github.com/Hashmi360/MRI_Recon_Foundation_Models
**Keywords:** Cardiac MRI Reconstruction, Foundation Models, U-Net

## 1. Introduction

Cardiac magnetic resonance (CMR) imaging, including multi-contrast techniques, is a vital clinical tool for comprehensive evaluation of cardiovascular diseases, offering detailed insights into cardiac structure and function. However, acquiring high-quality CMR images often involves lengthy scan durations, causing patient discomfort and motion artifacts that hinder reconstruction. Multi-contrast reconstruction from undersampled k-space further challenges the preservation of contrast-specific structural details (Xu et al., 2024).

Medical image reconstruction has seen significant progress through deep learning, particularly with convolutional neural networks (CNNs), especially U-Net-based architectures (Ronneberger et al., 2015; Lyu et al., 2025). However, these methods typically require extensive selection and tuning of hyperparameters and often fall short in preserving fine details or generalizing across diverse contrast types, limiting their adaptability in multi-contrast CMR reconstruction (Knoll et al., 2020).

Recently, vision foundation models trained on large-scale datasets have attracted considerable interest for their strong generalization and zero-shot capabilities across diverse tasks (Bommasani et al., 2021; Huix et al., 2024). Despite this progress, their potential for CMR reconstruction remains largely unexplored. In this work, we take a first step towards exploring the potential of vision foundation models for CMR reconstruction. **Our contributions are**: **(1)** To the best of our knowledge, this is the first study to investigate the use of vision foundation models, such as CLIP and DINOv2, for CMR reconstruction, exploring their transferability to this task; **(2)** We propose a novel reconstruction framework that fuses multi-level features from frozen image encoder of the foundation models using a trainable MLP and a UNETR-based decoder, complemented by a lightweight convolutional stem to enrich with domain-specific local feature details. **(3)** We evaluate CLIP and DINOv2 for CMR reconstruction on the CMRxRecon2024 dataset using NMSE, PSNR, and SSIM, comparing them against BiomedCLIP—a medical domain-specific model—and a U-Net trained from scratch.

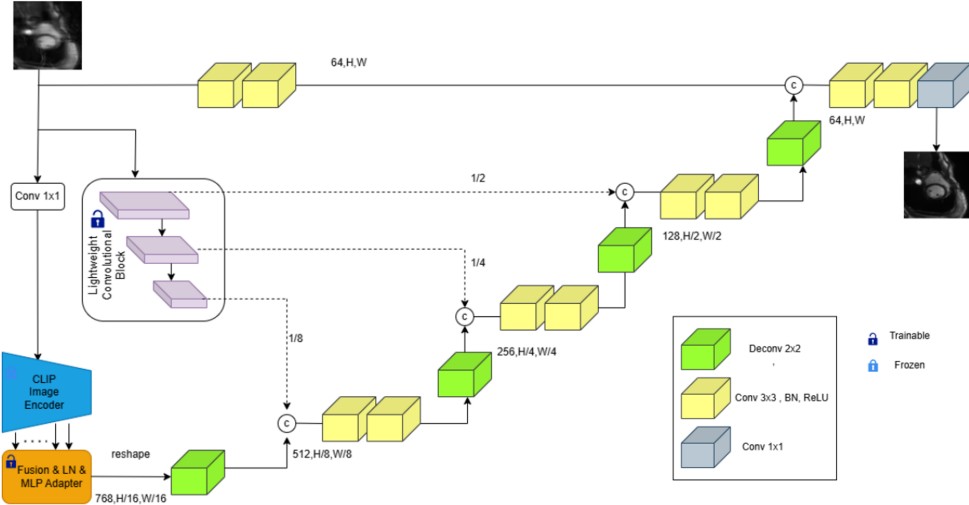

Figure 1: The architecture of proposed framework with frozen CLIP image encoder.

## 2. Method

Our framework, depicted in Figure 1, leverages a frozen CLIP (Radford et al., 2021) image encoder based on the ViT-B/16. To capture scale information effectively, we extract multi-level features from the image encoder's layers and concatenate them along the hidden dimension. These concatenated features are normalized using LayerNorm (Ba et al., 2016; Cai et al., 2024) for training stability and then processed through a trainable MLP layer. Additionally, we include a lightweight convolutional block (Qin et al., 2024) with three $3 \times 3$ convolutional layers, each followed by batch normalization and ReLU activation, to capture more local details. The outputs of this block are added as skip connections to the trainable UNETR-based decoder (Hatamizadeh et al., 2022), which integrates them to reconstruct images.

## 3. Experimental Results

We use the multi-contrast k-space data from the CMRxRecon2024 (Wang et al., 2025) challenge, including Cine, Aorta, Tagging, and Mapping contrasts, with 200 training samples split into 70% training, 10% validation, and 20% testing. Reconstruction performance is evaluated under acceleration factors AF ($\times 4, \times 8, \times 10$). In our experiments, we minimize SSIM loss between the target and reconstructed images. The input images have a resolution of $224 \times 224$, and a batch size of 8. Models are trained using the AdamW optimizer with a weight decay of 0.01 and an initial learning rate of $2 \times 10^{-4}$ on a GTX 4090 GPU.

The experimental results in Table 1, show that our CLIP-based framework outperforms the U-Net model trained from scratch at an AF of ($\times 4$), achieving a PSNR of 28.87 dB, and remains competitive across higher acceleration factors. Notably, CLIP surpasses both DINOv2 (Oquab et al., 2023) and the domain-specific BiomedCLIP (Zhang et al., 2023) model in overall performance. The trainable parameters for the models investigated are as follows: CLIP (20.87M), DINOv2 (20.92M), BiomedCLIP (20.87M) and U-Net (31.04M).

Table 1: Experimental Results on CMRxRecon2024 dataset

| Model | ($\times 4$) | | | ($\times 8$) | | | ($\times 10$) | | |
|---|---|---|---|---|---|---|---|---|---|
| | SSIM ↑ | PSNR ↑ | NMSE ↓ | SSIM ↑ | PSNR ↑ | NMSE ↓ | SSIM ↑ | PSNR ↑ | NMSE ↓ |
| CLIP ViT-B/16 | 0.8760 | **28.8750** | **0.0363** | 0.8259 | 26.8198 | 0.0576 | 0.8118 | 26.3439 | 0.0641 |
| DINOv2 ViT-B/14 | 0.8720 | 27.8272 | 0.0467 | 0.8243 | 26.7226 | 0.0595 | 0.8105 | 26.1954 | 0.0677 |
| BiomedCLIP ViT-B/16 | 0.8694 | 28.1403 | 0.0439 | 0.8224 | 26.6565 | 0.0611 | 0.8061 | 25.9521 | 0.0713 |
| U-Net | **0.8810** | 28.7364 | 0.0371 | **0.8357** | **27.1537** | **0.0534** | **0.8224** | **26.5216** | **0.0609** |

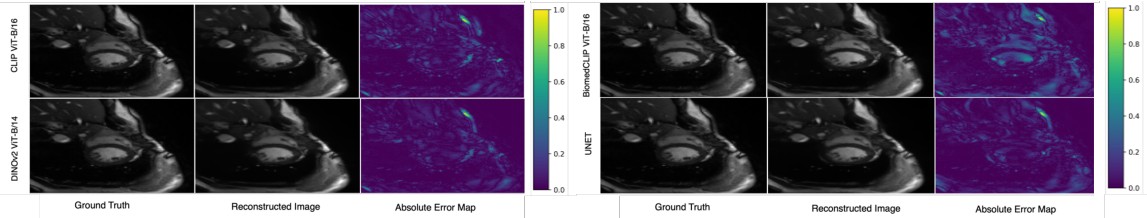

Figure 2: Reconstruction results and error maps for AF ($\times 4$), highlighting lower errors by the CLIP-based framework over U-Net and BiomedCLIP.

## 4. Conclusion

In this work, we explored the applicability of frozen vision foundation models, such as CLIP and DINOv2, for CMR reconstruction. While improvements—such as domain-specific fine-tuning are needed to fully optimize their performance, our findings underscore the promising potential of repurposing natural image foundation models for CMR reconstruction.

## Acknowledgments

This work was supported by Taighde Éireann – Research Ireland under Grant numbers 18/CRT/6183 and 12/RC/2289 P2.

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
