# OpenReview forum: "Can Natural Domain Foundation Models Be Applied to Cardiac MRI Reconstruction?"
_MIDL.io/2025/Short_Papers — MIDL 2025 - Short Papers_

### Official Review · Reviewer_hmEq · 2025-04-19

**Rating:** 3
**Confidence:** 4

**Summary:**

This paper attempts to address cardiac MRI reconstruction by mapping from zero-filled images to higher-quality reconstructions. It proposes to use pretrained natural image foundation models such as CLIP and DINOv2 as an encoder attached to a convolutional decoder. It performs experiments in the 4x, 8x, and 10x undersampled settings.

**Strengths:**

It is an interesting research question: to what extent are pretrained frozen features from a natural image model that has (likely) never seen undersampled zero-filled reconstructions of a heart able to aid in the unrelated task of MRI reconstruction?

**Weaknesses:**

If accepted, this submission should be reframed as a negative result.

- To my reading, the proposed method is worse than using a simple UNet across most undersampling factors and evaluation measures in Table 1 (7/9 settings).
- The only baseline in the paper (a simple image-to-image UNet without data consistency or a cascade, etc.) is not a standard baseline for CMRRecon. For example, the original CMRRecon data paper uses MoDL, GRAPPA, and SENSE as baselines for all three undersampling factors and the PSNR/SSIM is higher than what is reported here.
- There is no incorporation of MRI-specific considerations into the methodology. Some potential questions to consider in a full paper might be (1) would cascading or using data consistency help improve the reconstruction? (2) would using GRAPPA/SENSE reconstructions as input instead of zero-filling improve the reconstruction as they lessen the domain gap to natural images?

---

### Decision · Program_Chairs · 2025-05-01

Accept